# Black African men's perception of mental health in Africa: Scoping review protocol

**Lazarros Chavalala** *, **Lufuno Makhado**

Department of Public Health, University of Venda, Thohoyandou, South Africa

* lazarros2chavalala@gmail.com

## Abstract

Mental health is often viewed and interpreted differently from one society to the next. Black African men who live within African countries and those found in Western countries may view mental health differently and may shape the way issues impacting mental health are dealt with. The way mental health is promoted in an African perspective may differ from the way Western people promote it among their citizens, including men. Mental health is a crucial aspect of human daily functioning, and its absence can significantly impact one's daily functioning and interactions with others. This review aims to map available literature on Black African men's perception of mental health. This review will be guided by Arksey and O'Malley's five-stage framework incorporating PRISMA-ScR guidance. The search will be conducted on Google Scholar, Sabinet, Web of Science, PubMed, Organisational Websites, and CINAHL. Grey literature will also be collected from reports, University dissertations and theses from institutional repositories, as well as from organisations and government reports. Authors will independently screen and collect data from included studies and thematically analyse findings. The review expects to identify black African men's perception of mental health and categorise them based on themes that will emerge. The findings may be useful when formulating and revising health policies about men's mental health and can be used to identify gaps in men's mental health and ultimately inform interventions to improve men's mental health.

## Introduction

Mental health is an important part of men's holistic well-being and forms part of the intervention to improve the well-being of the general population worldwide [1]. The World Health Organisation (WHO) define mental health as a state of well-being in which every individual realises his or her own potential, can cope with the normal stresses of life, can work productively and fruitfully, and can contribute to his or her community [2].It is clear from the definition that mental health does not mean the absence of disease or illness, but being able to function and cope with mental illness or challenges that an individual is dealing with. There are various mental

**Data availability statement:** No datasets were generated or analyzed during the current study. All relevant data from this study will be made available upon study completion.

**Funding:** The author(s) received no specific funding for this work.

**Competing interests:** The authors have declared that no competing interests exist.

health disorders, but there are common ones. Common mental health disorders include post-traumatic stress disorder (PTSD), Generalised anxiety disorder, Major depressive disorder, Bipolar disorder, Manic episodes, and Hypomanic episodes [3,4]. Across the globe, mental disorders affect a large percentage of the population, with 12.7% of men experiencing mental health disorders, in contrast to 14.3% of females [5]. In the African region, the suicide rate is estimated at approximately 11.5 per 100,000 population, with men bearing a higher burden of suicide mortality [6]. A systematic review by Babajani et al [7] found that suicide attempt prevalence among African men was at 7.6%. Men have been seen as less likely to seek help in dealing with mental health problems and are more likely to take their own lives than women [8]. The men's unwillingness to seek help for mental health problems is problematic because mental health issues like loneliness and chronic stress can frequently be neglected in men. The mental and emotional health of men has been described as a 'silent crisis' [9]. Epidemiological studies of suicide, substance abuse, and depression in men reveal various shared underlying risk factors. A major shared risk factor is problems related to employment and occupation. Being unemployed can be a chronic stressor and greatly impact men's mental health compared to women [9]. On black African men, various factors affect their mental health and how mental health is dealt with. This includes masculinity norms rooted within the African cultures [10]. Within various African regions, masculinity is perceived differently, and the way it is understood varies from one region to another. In most communities, the idea of what it means to be a man is shaped by area customs, social institutions, colonial history, rites of passage, economic life and religion [11]. In West Africa, masculinity is perceived as having high decision-making power within the household and the community. This perception is rooted within the socialisation process that links masculinity with power, control and family welfare [12]. In East Africa, masculinity is perceived and centred around warriorhood, bravery, cattle ownership, and the successful completion of initiation rites marking the transition to manhood [13].In Southern Africa, masculinity has been reshaped by Apartheid, colonialism and labour migration. In this area, masculinity reinforces authority, heterosexual strength, toughness, and status as a provider, while simultaneously creating challenges and alternative, more equitable forms of masculinity [14].Considering how masculinity is perceived across the above African regions, it is very unlikely that men dealing with issues affecting mental health may easily come out, and many battle with mental health silently due to traditional norms and society expectations surrounding masculinity. In South Africa, we recognise the existence of movements promoting gender equality; numerous expectations held by individuals are still based on traditional gender, social, and cultural norms [15]. Though these norms can be a source of significant pride and strength, they can also create expectations that might negatively impact the mental health of individuals, regardless of gender [16]. In many African cultures, the cultural norms encourage men to act strongly and not present any symptoms of being weak when going through hardships, which may lead to the concealment of depression and increase the likelihood of not seeking help [17]. Conformity influencing men's involvement with psychological health services, conformity to traditional masculine norms

is believed to directly affect men's mental health conditions. Studies have found links between adherence to traditional masculine standards and depression, anxiety, aggressive behaviours, and other negative mental health effects [18–22]. A preliminary database search was conducted to determine if there were any previously conducted studies that mapped available literature on men's perception of mental health. We found that previously conducted scoping review studies focused on the general population and other population groups on their perceptions of mental health; no available published study collates evidence on Black African men's perceptions of mental health. For instance, a study by Meechan et al. (2021) [23] focused on black male adolescents living in the United Kingdom. Due to a lack of studies mapping available literature on Black African men's perception of mental health, the authors saw it important to close the gap and map available evidence on that will aid an understanding of how this population group perceive mental health.

## Materials and methods

### Study design

The study is guided by Arksey and O'Malley's five-stage framework [24]. Arksey and O'Malley's five-stage framework is a framework that gives guidance on conducting scoping reviews through five stage process, with stage 1 being Identifying the research question, followed by identifying relevant studies in stage 2. In stage 3, studies eligible for inclusion are selected, and data are extracted in stage 4. Once data collection is finished, the data are analysed, summarised, and the findings are reported [25]. The framework was chosen for its ability to provide a clear approach to mapping existing literature and to provide a transparent process for mapping literature.

### Aim

The review aims to map the degree of literature available and describe black African men's perception of mental health.

### Review question

The Population Concept Context (PCC) framework was applied to inform a research question that aligns well with the scoping review aim. The review question is: *What is the perception of black African men regarding mental health?*

### Search strategy

The literature search will be conducted on Google Scholar, Sabinet, Web of Science, PubMed, Organisational Websites, and CINAHL. The search target literature published between 2010 and 2025, Grey literature will also be collected from reports, University dissertations and theses from institutional repositories, organisations, and government reports will be used to search for grey literature. To ensure that the search strategy aligns with eligibility criteria, publication data range, language and publication type filters will be applied when searching for literature. The search strategy has been developed using a combination of keywords and Boolean operators. The search structure on databases is also presented in Table 1. Below are the search terms and Boolean operators to be used.

(Black African Men OR African men OR African males)

AND

("Mental health" OR "psychological health" OR "psychological wellbeing" OR "cognitive well-being")

AND

(perception OR understanding OR interpretation OR Views)

**Table 1. Search structure on databases.**

| Database | Search structure |
|---|---|
| **PubMed** | ("viewed"[All Fields] OR "viewing"[All Fields] OR "viewings"[All Fields] OR "views"[All Fields] OR ("percept"[All Fields] OR "perceptibility"[All Fields] OR "perceptible"[All Fields] OR "perception"[MeSH Terms] OR "perception"[All Fields] OR "perceptions"[All Fields] OR "perceptional"[All Fields] OR "perceptive"[All Fields] OR "perceptiveness"[All Fields] OR "percepts"[All Fields]) OR ("comprehension"[MeSH Terms] OR "comprehension"[All Fields] OR "understand"[All Fields] OR "understanding"[All Fields] OR "understands"[All Fields] OR "understandability"[All Fields] OR "understandable"[All Fields] OR "understandably"[All Fields] OR "understandings"[All Fields]) OR ("interpret"[All Fields] OR "interpretability"[All Fields] OR "interpretable"[All Fields] OR "interpretating"[All Fields] OR "interpretation"[All Fields] OR "interpretation s"[All Fields] OR "interpretational"[All Fields] OR "interpretations"[All Fields] OR "interpretative"[All Fields] OR "interpreted"[All Fields] OR "interpreter"[All Fields] OR "interpreter s"[All Fields] OR "interpreters"[All Fields] OR "interpreting"[All Fields] OR "interpretive"[All Fields] OR "interpretively"[All Fields] OR "interprets"[All Fields])) AND ("Black African Men"[All Fields] OR "African men"[All Fields] OR "African males"[All Fields] OR "African man"[All Fields]) AND ("Mental health"[All Fields] OR "psychological health"[All Fields] OR "cognitive well-being"[All Fields] OR "psychological wellbeing"[All Fields]) |
| **Sabinet** | ("Black African Men" OR "African men" OR "African males" AND "Mental health" OR "psychological health" OR "cognitive well-being" OR psychological Wellbeing AND perception OR understanding OR interpretation OR Views) |
| **Web of Science** | ("Black African Men" OR "African men" OR "African males" AND "Mental health" OR "psychological health" OR "cognitive well-being" OR psychological Wellbeing AND perception OR understanding OR interpretation OR Views) Query builder will be used to combine search results and apply filters for publication timeline, discipline and document type. |
| **Google scholar** | ("Black African Men" OR "African men" OR "African males" AND "Mental health" OR "psychological health" OR "cognitive well-being" OR psychological Wellbeing AND perception OR understanding OR interpretation OR Views) A hand search will be performed, applying the Boolean operators to screen the first relevant 400 articles. |
| **CINAL** | ("Black African Men" OR "African men" OR "African males" AND "Mental health" OR "psychological health" OR "cognitive well-being" OR psychological Wellbeing AND perception OR understanding OR interpretation OR Views) |

## Eligibility criteria and study selection

We will use eligibility criteria to select studies on the black African men's perception of mental health (Table 2).

## Inclusion criteria

- Studies reporting Black African men's perception of mental health
- Articles published between 2010 and 2025
- Articles published in the English language

## Exclusion criteria

Studies reporting perception of women, children and other population groups other than black African men will be excluded.

**Table 2. Framework for determining the eligibility of the research question.**

| Population | Human participants, Black African adult males aged 18 years and older, Reside within African countries |
|---|---|
| Concept | Perception of Mental Health. |
| Context | Studies conducted between 2010 and 2025 within the African continent |

Studies conducted on men of other races

Studies conducted on Black men residing outside the African countries

Studies reporting clinical outcomes of men's mental disorders without exploring perceptions.

## Study section

Study selection will follow PRISMA-ScR guidelines attached as S1 PRISMA Checklist to reveal transparency on identification, screening, eligibility, inclusion and details of included sources and synthesised findings [26]. All identified sources will be imported into Rayyan (an online systematic review management platform) for deduplication. The authors will then independently screen titles and abstracts against the inclusion criteria in an exported Excel spreadsheet. Sources that will be deemed relevant will be fully screened by the two authors independently, and disagreements will be addressed through dialogue. A third reviewer will be sought to intervene and address any disagreement when the two authors fail to address their disagreement. Sources that will be excluded from full screening will be documented with reasons for exclusion in the final review report. The Prisma flow diagram will be used to illustrate the process of selecting, screening, eligibility, and inclusion [27].

## Data charting and management

The two reviewers will independently collect data from included studies using a standardised data collection form. The following data elements will be collected from the included studies:

• Author (s) and year of publication

• Study location,

• Aim

• Study design.

• Population characteristics

• Findings on perception of mental health.

The quality of included studies/sources will be appraised using the Mixed Methods Appraisal Tool (MMAT) [28], as it allows appraisal of qualitative, quantitative, and mixed methods designs. Collected data will be inductively thematically analysed and put into themes based on reported perceptions. Once data has been collected, reviewers will manually go through the collected data and read it to familiarise themselves with the data. Then, codes will be generated from the data, the generated codes will be grouped based on their similarities, and themes will be formulated. To ensure intercoder reliability, a third independent reviewer will be appointed to validate the developed themes. The reviewer will closely look at how codes were generated, categorised and how themes were generated to ensure consistency and confirm if the developed themes reflect the data accurately. Differences and disagreements that may arise will be addressed dialogues

until a consensus is reached. Due to the expected heterogeneity of study designs, meta-analysis will not be performed; hence, findings will be narratively synthesised.

## Ethical considerations

This study does not involve human subjects; therefore, it does not require ethical approval. However, the protocol has been registered at Open Science Framework (OSF), and registration details are available at DOI 10.17605/OSF.IO/8QSE4.

## Discussion

This scoping review protocol gives a plan for synthesising evidence on Black African male perception of mental health in the literature published between 2010 and 2025. The PCC framework has been used to guide the formulation of relevant review questions for the scoping review. It plans to employ PRISMA-ScR guidelines for transparency in its methods, from identifying to the inclusion of sources. It also plans to use the Mixed Methods Appraisal Tool (MMAT) to appraise studies to be included. The findings of this review are expected to inform health policy, strengthen interventions aimed at improving men's mental health, and identify gaps within men's mental health which may be further researched.

## Strengths and limitations

This review's strength is that it applies rigorous methods, including Arksey and O'Malley framework and Prisma guidelines, which strengthen its trustworthiness and transparency on procedures that will be followed during the study. The use of the PCC framework also enhances transparency on how the review question was formulated and ensures that relevant and comprehensive literature is sought in different databases. Independent dual review by the two authors strengthens the reliability of the methods to be used to conduct the review. However, limitations of this review include a restriction to the English language; thus, articles and reports that report men's perceptions in other languages rather than English will be missed, even if they contain the required data. Moreover, emerging evidence outside the study's focus time frame (2010–2025) will be missed.

### Plans for dissemination

The review findings will be submitted to a peer-reviewed open-access journal for publication to ensure accessibility. The findings will also be shared with stakeholders, health institutions and policy makers. The findings will further be presented at professional and academic conferences on men's health scheduled for 2026.

### Status and timeline

The review will commence on 15 December 2025 using the methods and following procedures provided in this protocol. Any changes made during review will be documented. The time frame presented in Table 3 is the estimated time frame of tasks the review will likely take (Table 3).

**Table 3. Review completion timeline.**

| Activity | Time frame |
|---|---|
| Protocol development | November 2025 |
| Searching and selection of eligible studies | December 2025-January 2026 |
| Data extraction, Harting and critical appraisal | February 2026-March 2026 |
| Analysis, results and synthesis | April to May 2026 |

## Supporting information

**S1 PRISMA Checklist. Completed PRISMA-ScR checklist for the scoping review.** This material is distributed under the Creative Commons Attribution 4.0 International License (CC BY 4.0), which permits unrestricted use, distribution, and reproduction provided the original authors and source are credited.
(PDF)

## Author contributions

**Conceptualization:** L Chavalala, Lufuno Makhado.

**Methodology:** L Chavalala, Lufuno Makhado.

**Validation:** L Chavalala, Lufuno Makhado.

**Writing – original draft:** L Chavalala, Lufuno Makhado.

**Writing – review & editing:** L Chavalala, Lufuno Makhado.

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
