## [Decision Letter · Decision Letter 0]

17 Feb 2026

PMEN-D-25-00574

Black African men’s perception of mental health in the African continent: Scoping review protocol

PLOS Mental Health

Dear Dr. Chavalala,

Thank you for submitting your manuscript to PLOS Mental Health. After careful consideration, we feel that it has merit but does not fully meet PLOS Mental Health’s publication criteria as it currently stands. Therefore, we invite you to submit a revised version of the manuscript that addresses the points raised during the review process.

We look forward to receiving your revised manuscript.

Kind regards,

Lambert Zixin Li, Ph.D.

Academic Editor

PLOS Mental Health

Journal Requirements:

1. We noticed you have some minor occurrence of overlapping text with the following previous publication(s), which needs to be addressed:

- https://doi.org/10.1177/0706743718762388

- https://www.safmh.org/a-silent--pandemic/

In your revision ensure you cite all your sources (including your own works), and quote or rephrase any duplicated text outside the methods section. Further consideration is dependent on these concerns being addressed.

2. We have noticed that you have uploaded Supporting Information files, but you have not included a list of legends. Please add a full list of legends for your Supporting Information files after the references list.

Additional Editor Comments (if provided):

Thank you for submitting your manuscript. After careful consideration of the reviewers’ comments, we invite you to revise and resubmit your paper.

The reviewers have raised substantial methodological concerns that affect the clarity, rigor, and validity of the study, particularly with respect to the study design and the level of evidence that can be drawn from it. A thorough revision addressing these issues will be necessary before the manuscript can be reconsidered.

Please provide a detailed, point-by-point response to the reviewers’ comments and clearly indicate how the manuscript has been revised.

We look forward to receiving your revised submission.

Reviewers' comments:

Reviewer's Responses to Questions

**Comments to the Author**

1. Does this manuscript meet PLOS Mental Health’s publication criteria? Is the manuscript technically sound, and do the data support the conclusions? The manuscript must describe methodologically and ethically rigorous research with conclusions that are appropriately drawn based on the data presented.? Is the manuscript technically sound, and do the data support the conclusions? The manuscript must describe methodologically and ethically rigorous research with conclusions that are appropriately drawn based on the data presented.

Reviewer #1: Yes

Reviewer #2: No

Reviewer #3: Yes

2. Has the statistical analysis been performed appropriately and rigorously?

Reviewer #1: Yes

Reviewer #2: I don't know

Reviewer #3: N/A

3. Have the authors made all data underlying the findings in their manuscript fully available (please refer to the Data Availability Statement at the start of the manuscript PDF file)?

The PLOS Data policy requires authors to make all data underlying the findings described in their manuscript fully available without restriction, with rare exception. The data should be provided as part of the manuscript or its supporting information, or deposited to a public repository. For example, in addition to summary statistics, the data points behind means, medians and variance measures should be available. If there are restrictions on publicly sharing data—e.g. participant privacy or use of data from a third party—those must be specified.requires authors to make all data underlying the findings described in their manuscript fully available without restriction, with rare exception. The data should be provided as part of the manuscript or its supporting information, or deposited to a public repository. For example, in addition to summary statistics, the data points behind means, medians and variance measures should be available. If there are restrictions on publicly sharing data—e.g. participant privacy or use of data from a third party—those must be specified.

Reviewer #1: Yes

Reviewer #2: No

Reviewer #3: Yes

4. Is the manuscript presented in an intelligible fashion and written in standard English?

Reviewer #1: Yes

Reviewer #2: No

Reviewer #3: Yes

Reviewer #1: 1. The protocol does not clearly discuss how "African masculinity" is understood in different sub-regions. This could lead to inaccurate generalizations.

2. The criteria for "Black African men" need to be clearer. It should be specified whether this includes individuals in the diaspora or only those living in Africa.

3. The plan for thematic analysis is not specific. The authors should indicate whether they will use inductive or deductive coding.

4. It is important to state clearly whether the review includes studies of Black African men in Western countries or is only about those on the African continent, as the title suggests.

5. When discussing mental health intervention strategies. It is suggested to refer from the paper: “A Systematic Review of Mental Health Monitoring and Intervention Using Unsupervised Deep Learning on EEG Data “. Including this suggested reference adds a modern view on mental health monitoring and intervention techniques. It strengthens the manuscript by showing advances in data-driven mental health practices, which contrast with the current focus on perceptions and qualitative methods.

6. The authors should explain what the "five-stage framework" is and how they will validate the themes, for example, through a third reviewer or software such as NVivo.

7. The term "Black African men" should be used consistently throughout the document, avoiding variations like "African men" or "men in the African context."

8. While the protocol cites important works, it should also include recent systematic reviews on African mental health to provide a stronger foundation for the study.

Reviewer #2: Great topic with an excellent goal of improving policies on mental health for black African men. However, manuscript was presented as a research article but lacks the depth of information to substantiate the claim of a research article. It also needs more key words and improved grammatical sentences.

Since it's in a scoping review protocol phase, perhaps it would be better to present the research article for publication review after the final stage of analysis, results and synthesis (April-May)

This will give the authors the chance to perform more in-depth analysis which also improves the depth of information for meeting the study's goal/purpose

Thank you for the invitation to review this manuscript.

Reviewer #3: I have very few recommendations for you:

Study Design:

Mention the steps of the scoping review, the one you followed in this paper Arksey O'Maley and give Headlines accordingly.

Inclusion and Exclusion Criteria:

We do not include the points in exclusion criteria which are already mentioned in the inclusion criteria. For example, the review includes english language literatures. You do not need to mention it in the exlusion criteria. Instead add points that are not in the inclusion criteria. You need to change the exclusion criteria completely. For instance, you can add- Articles related to mental health for women/girls in the study. I am giving you just examples to make it easier for you to understand.

**Do you want your identity to be public for this peer review?** For information about this choice, including consent withdrawal, please see our Privacy Policy..

Reviewer #1: No

Reviewer #2: **Yes:**Nonye Tochi Aghanya, MSc, RN, FNP-CNonye Tochi Aghanya, MSc, RN, FNP-CNonye Tochi Aghanya, MSc, RN, FNP-CNonye Tochi Aghanya, MSc, RN, FNP-C

Reviewer #3: **Yes:**Rehnuma AbdullahRehnuma AbdullahRehnuma AbdullahRehnuma Abdullah

---

## [Decision Letter · Decision Letter 1]

12 Mar 2026

Black African men’s perception of mental health in  African: Scoping review protocol

PMEN-D-25-00574R1

Dear Dr Chavalala,

We are pleased to inform you that your manuscript 'Black African men’s perception of mental health in  African: Scoping review protocol' has been provisionally accepted for publication in PLOS Mental Health.

Best regards,

Lambert Zixin Li, Ph.D.

Academic Editor

PLOS Mental Health

Dear Authors,

Thank you for submitting your manuscript to PLOS Mental Health. After consideration of the reviews and your revisions, I am pleased to inform you that the manuscript is accepted for publication. The reviewers’ comments have been carefully addressed, and the paper makes a valuable contribution to the field.

Congratulations, and thank you for choosing the journal for your work.

Best regards,

Lambert Zixin Li, PhD

Reviewer Comments (if any, and for reference):

Reviewer's Responses to Questions

**Comments to the Author**

Reviewer #1: All comments have been addressed

Reviewer #2: All comments have been addressed

publication criteria? Is the manuscript technically sound, and do the data support the conclusions? The manuscript must describe methodologically and ethically rigorous research with conclusions that are appropriately drawn based on the data presented.? Is the manuscript technically sound, and do the data support the conclusions? The manuscript must describe methodologically and ethically rigorous research with conclusions that are appropriately drawn based on the data presented.

Reviewer #1: Yes

Reviewer #2: Yes

3. Has the statistical analysis been performed appropriately and rigorously?

Reviewer #1: Yes

Reviewer #2: I don't know

4. Have the authors made all data underlying the findings in their manuscript fully available (please refer to the Data Availability Statement at the start of the manuscript PDF file)?

The PLOS Data policy requires authors to make all data underlying the findings described in their manuscript fully available without restriction, with rare exception. The data should be provided as part of the manuscript or its supporting information, or deposited to a public repository. For example, in addition to summary statistics, the data points behind means, medians and variance measures should be available. If there are restrictions on publicly sharing data—e.g. participant privacy or use of data from a third party—those must be specified.requires authors to make all data underlying the findings described in their manuscript fully available without restriction, with rare exception. The data should be provided as part of the manuscript or its supporting information, or deposited to a public repository. For example, in addition to summary statistics, the data points behind means, medians and variance measures should be available. If there are restrictions on publicly sharing data—e.g. participant privacy or use of data from a third party—those must be specified.

Reviewer #1: Yes

Reviewer #2: Yes

5. Is the manuscript presented in an intelligible fashion and written in standard English?

Reviewer #1: Yes

Reviewer #2: Yes

**Reviewer #1:**No further major recomemndations No further major recomemndations No further major recomemndations No further major recomemndations

**Reviewer #2:**A more refined version of a very important topic worthy of publication. Author addressed reviewer recommendations: Increased key words count, more clearly defined interpretation of "African masculinity" in various African regions with clearer inclusion and exclusion of study criteria. There was also a much better explanation of the study design, Arksey and O'Malley five stage frameworkA more refined version of a very important topic worthy of publication. Author addressed reviewer recommendations: Increased key words count, more clearly defined interpretation of "African masculinity" in various African regions with clearer inclusion and exclusion of study criteria. There was also a much better explanation of the study design, Arksey and O'Malley five stage frameworkA more refined version of a very important topic worthy of publication. Author addressed reviewer recommendations: Increased key words count, more clearly defined interpretation of "African masculinity" in various African regions with clearer inclusion and exclusion of study criteria. There was also a much better explanation of the study design, Arksey and O'Malley five stage frameworkA more refined version of a very important topic worthy of publication. Author addressed reviewer recommendations: Increased key words count, more clearly defined interpretation of "African masculinity" in various African regions with clearer inclusion and exclusion of study criteria. There was also a much better explanation of the study design, Arksey and O'Malley five stage framework

**Do you want your identity to be public for this peer review?** For information about this choice, including consent withdrawal, please see our Privacy Policy..

Reviewer #1: No

Reviewer #2: **Yes:**Nonye Tochi Aghanya MSc, RN, FNP-CNonye Tochi Aghanya MSc, RN, FNP-CNonye Tochi Aghanya MSc, RN, FNP-CNonye Tochi Aghanya MSc, RN, FNP-C
